# Development of a Novel Quinoline Derivative as a P-Glycoprotein Inhibitor to Reverse Multidrug Resistance in Cancer Cells

**DOI:** 10.3390/biology8040075

**Published:** 2019-10-01

**Authors:** Yuanyuan Zhou, Po-yee Chung, Jessica Yuen-wuen Ma, Alfred King-yin Lam, Simon Law, Kwok-wah Chan, Albert Sun-chi Chan, Xingshu Li, Kim-hung Lam, Chung-hin Chui, Johnny Cheuk-on Tang

**Affiliations:** 1State Key Laboratory of Chemical Biology and Drug Discovery, Lo Ka Chung Centre for Natural Anticancer Drug Development, Department of Applied Biology and Chemical Technology, The Hong Kong Polytechnic University, Hong Kong 999077, China; yuanyuan.09.zhou@connect.polyu.hk (Y.Z.);; 2School of Optometry, The Hong Kong Polytechnic University, Hong Kong 999077, China; majessicayuenwuen@gmail.com; 3Griffith Medical School, Griffith University, Gold Coast, QLD 4222, Australia; A.Lam@griffith.edu.au; 4Department of Surgery, Li Ka Shing Faculty of Medicine, The University of Hong Kong, Hong Kong 999077, China; slaw@hku.hk; 5Department of Pathology, Li Ka Shing Faculty of Medicine, The University of Hong Kong, Hong Kong 999077, China; kwchan@pathology.hku.hk; 6School of Pharmaceutical Sciences, Sun Yat-sen University, Guangzhou 510006, China; chenxz3@mail.sysu.edu.cn (A.S.-c.C.); lixsh@mail.sysu.edu.cn (X.L.)

**Keywords:** anticancer, multidrug resistance, p-glycoprotein, quinoline compounds

## Abstract

Multidrug resistance (MDR) is one of conventional cancer chemotherapy’s limitations. Our group previously synthesized a series of quinoline-based compounds in an attempt to identify novel anticancer agents. With a molecular docking analysis, the novel compound 160a was predicted to target p-glycoprotein, an MDR candidate. The purpose of this study is to evaluate 160a’s MDR reversal effect and investigate the underlying mechanism at the molecular level. To investigate 160a’s inhibitory effect, we used a series of parental cancer cell lines (A549, LCC6, KYSE150, and MCF-7), the corresponding doxorubicin-resistant cell lines, an MTS cytotoxicity assay, an intracellular doxorubicin accumulation test, and multidrug resistance assays. The Compusyn program confirmed, with a combination index (CI) value greater than 1, that 160a combined with doxorubicin exerts a synergistic effect. Intracellular doxorubicin accumulation and transported calcein acetoxymethyl (AM) (a substrate for p-glycoprotein) were both increased when cancer cells with MDR were treated with compound 160a. We also showed that compound 160a’s MDR reversal effect can persist for at least 1 h. Taken together, these results suggest that the quinoline compound 160a possesses high potential to reverse MDR by inhibiting p-glycoprotein-mediated drug efflux in cancer cells with MDR.

## 1. Introduction

Cancer continues to be the most threatening disease in the world, and the development of anticancer drugs with high efficacy and minimal side effects remains a challenge for both academia and the pharmaceutical industry [1]. The development of drug resistance during chemotherapy seriously limits the survival rate of cancer patients [2]. Multidrug resistance (MDR) in cancer cells is a phenomenon that manifests as cross-resistance to a variety of structurally and mechanistically unrelated anticancer drugs. MDR renders cancer cells immune to standard treatments and is a major limitation of cancer therapy [3]. Drug resistance can occur in many ways, including altering the metabolism of a chemotherapeutic drug to make cells resistant to it.

The development of strategies to overcome MDR in cancer cells is a key challenge for effectively targeting the drug delivery system for successful chemotherapy. There are many potential MDR candidates, including the ABC transporter family, apoptosis, autophagy, cancer stem cell regulation, micro-RNA (miRNA) regulation, hypoxia, DNA damage and repair, and epigenetic regulation [4]. The ABC transporter family has at least 48 members, including P-glycoprotein, which is encoded by the ABCB1 gene, MDR-associated protein 1, which is encoded by the ABCC1 gene, and breast cancer resistance protein (BCRP), which is encoded by the ABCG2 gene. ABC transporters were found to be overexpressed on the membranes of cells with MDR, which increased the drug efflux, resulting in resistance to chemotherapy treatment [5]. To overcome multidrug resistance in cancer treatment, many strategies have been adopted, including chemotherapy with a combination of anti-tumor drugs and a P-gp (p-glycoprotein) inhibitor, inactivation of the MDR-associated gene by the targeting of specific mRNA, and development of a nanocarrier drug delivery system [6,7].

P-gp belongs to the first member of the ATP-binding cassette (ABC) superfamily, which is encoded by the MDR1 gene [8], and is widely expressed in epithelial cells of normal tissues that are involved in drug disposition, including the liver, intestine, and kidney [9]. It has been reported that P-gp (also called ABCB1 or MDR1) can actively bind and pump out drugs against the concentration gradient by hydrolysis of ATP and is an ATP-powered drug efflux pump membrane transporter; therefore, anticancer drugs cannot accumulate inside the cells to trigger cytotoxic effects. Some commonly used anticancer drugs have been shown to be weakened by P-gp, including vincristine, etoposide, and doxorubicin [10].

The first generation of P-gp inhibitors includes verapamil, trifluoperazine, cyclosporine-A, quinidine, reserpine, vincristine, yohimbine, tamoxifen, and toremifene, which are all competitive substrates of P-glycoprotein [11]. The second generation of P-gp inhibitors includes dexverapamil, dexniguldipine, valspodar (PSC 833), and biricodar citrate (VX-710), which are structurally modified first-generation P-gp inhibitors with reduced toxicity and better tolerability [12]. The third generation of P-gp inhibitors includes tariquidar-XR9576, zosuquidar-LY335979, laniquidar-R101933, and ONT-093, which have higher potency and specificity for P-gp than the second-generation P-gp inhibitors [13].

In this study, a quinoline derivative, 160a, was proven by our group to be capable of reversing the MDR phenotype in P-gp-overexpressing tumor cells. It has also been shown that the chloroquine resistance of *Plasmodium falciparum* malaria parasites has many similarities to MDR in tumor cells [14], suggesting that the relevant quinoline-based compounds may have potential to be novel anti-MDR agents. The results of this study may help to pave the path to the future development of novel anti-MDR agents against cancers.

## 2. Materials and Methods

### 2.1. Synthesis of Compound 160a

Compound 160a (8-(3-methoxybenzyloxy) quinoline-2-carbaldehyde) was synthesized through oxidation of 8-(3-methoxybenzyloxy)-2-methylquinoline by Dr. Penny Chan Sau-hing from our research group. Briefly, compound 160a was prepared by oxidation of 8-(3-methoxybenzyloxy)-2-methylquinoline with addition of selenium dioxide, pre-dried 1,4-dioxane and water in reflux for 24 h. 8-(3-Methoxybenzyloxy)-2-methylquinoline was synthesized by nucleophilic substitution of commercially available 2-methyl-8-quinolinol with 3-methoxybenzyl bromide in DMF at room temperature. Compound 160a was completely dissolved in dimethyl sulfoxide (DMSO) and, in this project, we examined its MDR-reversing effect on cancer cells in vitro and the underlying mechanisms. The structure of compound 160a was examined using 1H-NMR and ultra performance liquid chromatography/mass–mass spectrometry (UPLC/MS-MS; see Appendix A). Figure 1 shows the structure of compound 160a.

### 2.2. Cell Lines and Cell Culture

A total of nine cell lines were used to evaluate for the effect of the test compounds in this study. The human breast cancer cell line (LCC6 [15]) was kindly provided by Prof. Larry Chow from the Department of Applied Biology and Chemical Technology, Hong Kong Polytechnic University. The esophageal squamous cell carcinoma cell line (KYSE150 [16]) was purchased from Deutsche Sammlung van Mikroorganismen und Zellkulturen, Braunschweig (DSMZ, Braunschweig, Germany). The lung cancer cell line (A549) and the metastatic breast cancer cell line (MCF-7) were purchased from the American Type Culture Collection (ATCC, Manassas, VA, USA). The non-cancer esophageal epithelial cell line (NE-3, [17]) was kindly provided by Professor George S. W. Tsao from the Department of Anatomy at the University of Hong Kong. The LCC6/MDR cell line (an LCC6 cell line with multi-drug resistance [15]) and the MX100 cell line (an MCF-7 cell line with doxorubicin resistance) were kindly provided by Prof. Larry Chow. Two doxorubicin-resistant cell lines, DOX-KYSE150 and DOX-A549, were obtained from the parental cell lines (KYSE150 and A549) via culturing in an increasing concentration of doxorubicin (Sigma-Aldrich, Louis, MO, USA) starting from 0.1 μg/mL according to the previous report [18] with minor modifications. Surviving cells were repeatedly subcultured in a medium containing an increasing concentration of doxorubicin (0.1 μg/mL, 0.2 μg/mL, 0.5 μg/mL, 0.75 μg/mL, 1.00 μg/mL).

The A549, LCC6, LCC6/MDR, MCF-7, and MX100 cell lines were cultured in Dulbecco’s Modified Eagle’s Medium (Gibco, Carlsbad, CA, USA) and supplemented with 10% heat-inactivated fetal bovine serum (Biosera, Nuaille, France) and 100 units/mL penicillin (Gibco, USA). KYSE150 cells were cultured in 90% RPMI (Roswell Park Memorial Institute, Buffalo, NY, USA) 1640 medium and supplemented with 10% heat-inactivated fetal bovine serum and 100 units/mL penicillin. DOX-KYSE150 cells were cultured in 90% RPMI medium and supplemented with 10% heat-inactivated fetal bovine serum, 100 units/mL penicillin, and 1.00 μg/mL of doxorubicin. The DOX-A549 cell line was cultured in 90% DMEM medium and supplemented with 10% heat-inactivated fetal bovine serum, 100 units/mL penicillin, and 1.00 μg/mL of doxorubicin.

The cultures were maintained in a humidified atmosphere of 95% air and 5% CO_2_ at 37 °C. The cultures were passaged at pre-confluent densities of approximately 80% using 0.25% trypsin. Cells were washed briefly with phosphate-buffered saline (PBS), treated with 0.25% trypsin, and harvested for subculturing [19].

### 2.3. Molecular Docking Analysis

Prediction of the selected compound’s molecular binding targets was performed using the similarity ensemble approach (SEA) and the search engine http://sea.bkslab.org [20]. The binding behavior of compound 160a, or a positive control including doxorubicin and verapamil, to protein targets was investigated based on their molecular structures, which were matched against the ChEMBL medicinal chemistry database (version16). DockingServer (http://www.dockingserver.com/web) was used to determine the binding affinity of the compound to its predicted target and compare it to the binding affinity of the protein’s natural ligand by estimating the released free energy of the binding reactions.

### 2.4. Cytotoxicity Assay

A 3-(4,5-Dimethylthiazol-2-yl)-5-(3-carboxymethoxyphenyl)-2-(4-sulfophenyl)-2H-tetrazolium (MTS) assay was performed to evaluate the cytotoxic effect of compound 160a, doxorubicin, and a combination of compound 160a and doxorubicin on selected cell lines by using CellTiter96 AQueous One Solution Cell Proliferation (Promega, Madison, WI, USA) following the manufacturer’s instructions [21]. Briefly, approximately 5 × 10^3^ cells were seeded into each well of a flat-bottom 96-well cell culture plate in 100 μL of recommended culture medium and were allowed to grow for 24 h at 37 °C with 5% CO_2_ to settle down. After 24 h of incubation, the old culture medium was replaced by fresh medium with the treated test agents. The concentrations of doxorubicin were gradually increased from 0.000 μg/mL to 1.000 μg/mL (0.000, 0.031, 0.062, 0.125, 0.250, 0.500, 1.000 μg/mL). The concentrations of compound 160a in the 160a and doxorubicin combination cytotoxicity test were 5 μg/mL and 10 μg/mL. The concentrations of compound 160a in the cytotoxicity test were gradually increased from 0.000 μg/mL to 20 μg/mL (0.000, 0.625, 1.250, 2.500, 5.000, 10.000, 20.000 μg/mL) and 0.1% DMSO was added to the medium as a vehicle control (*n* = 4). The seeded 96-well plates were then incubated for 48 h at 37 °C with 5% CO_2_. The results were analyzed by measuring the absorbance at 492 nm using a microplate reader (Bio-RAD, Ultrmark, Microplate Imaging System, Hercules, CA, USA) to determine cell viability. The control value corresponding to untreated cells was taken to be 100%, and the viability of treated samples is expressed as a percentage of the control value.

### 2.5. Cell Proliferation Assay

A 3-(4,5-Dimethylthiazol-2-yl)-5-(3-carboxymethoxyphenyl)-2-(4-sulfophenyl)-2H-tetrazolium (MTS) assay was performed to analyze the effect of doxorubicin, compound 160a, and a combination of compound 160a and doxorubicin on cell proliferation in the dox-resistant cell lines (A549-DOX, KYSE150-DOX, MX100, and LCC6 MDR) as previously reported [22] with minor modifications. Approximately 5000 cells were seeded into each well of a flat-bottom 96-well plate in 100 μL of appropriate culture medium. After incubation at 37 °C in 5% CO_2_ overnight, the culture medium was replaced by a proper culture medium of 200 μL of 10 μg/mL compound 160a, 0.15 μg/mL doxorubicin, or 10 μg/mL compound 160a combined with 0.15 μg/mL doxorubicin (for MX100, the compound concentrations were 10 μg/mL compound 160a, 0.15 μg/mL doxorubicin, and 10 μg/mL compound 160a combined with 0.05 μg/mL doxorubicin). A 0.1% DMSO culture medium was used as a vehicle control group. MTS solution was added to each well after the culture medium was removed at 0 h, 6 h, 24 h, 48 h, and 72 h, respectively, without disturbing the attached cells, and then incubated at 37 °C in 5% CO_2_ for 45 min for the A549 and A549-DOX cell lines, 1 h for the LCC6 and LCC6 MDR cell lines, and 2 h for the MCF-7, MX100, KYSE150, and KUSE150-DOX cell lines. The absorbance of each well was measured at 492 nm using a microplate reader (Bio-RAD, Ultrmark, Microplate Imaging System, USA) to determine cell viability. The relative growth (compared with the cell viability at 0 h) of each cell line was then calculated using the equation:$$\frac{{\left[A\right]}_{T}}{{\left[A\right]}_{T0}},$$
where [A]_T_ is the absorbance at time point T and [A]_T0_ is the absorbance at 0 h. This assay was performed in triplicate.

### 2.6. Multidrug Resistance Analysis

A Multidrug Resistance Assay Kit (Cayman Chemical, Ann Arbor, MI, USA) was used to analyze the modulation of cellular MDR machinery. Calcein AM is a p-glycoprotein substrate. Compound 160a’s effect on p-glycoprotein can be evaluated using the intracellular calcein AM concentration as detected by flow cytometry and/or confocal microscopy.

Approximately 8 × 10^5^ cells were seeded into each well of a six-well plate and grown for 24 h at 37 °C in 5% CO_2_ to allow the cells to settle down. After the 24-h incubation, the cells were rinsed once with PBS and 1 mL of the culture medium was added with a gradually increasing concentration (2, 5, and 10 μg/mL) of compound 160a (6.82 μM, 17.06 μM, and 34.12 μM, respectively), verapamil (provided by the kit and diluted to 1:1000 in an appropriate medium) as a positive control, and 0.05% DMSO as a negative control (*n* = 3). Then, we incubated the treated cells at 37 °C in 5% CO_2_ and added calcein AM (provided by the kit). After incubation, cells were harvested and analyzed using the BD Accuri C6 flow cytometry system (BD Biosciences, San Jose, CA, USA).

Intracellular calcein AM accumulation was also observed using a confocal microscope. Approximately 8 × 10^5^ cells were seeded in glass-bottom microwell dishes (MatTek, Ashland, MA, USA) and allowed to grow for 24 h at 37 °C in 5% CO_2_. After the 24-h incubation, the cells were rinsed once with PBS and 1 mL of the culture medium was added with an increasing concentration (2, 5, and 10 μg/mL) of compound 160a (6.82 μM, 17.06 μM, and 34.12 μM, respectively), cyclosporin A (provided by the kit and diluted to 1:1000 in an appropriate medium) as the positive control, and 0.05% DMSO as the negative control. Then, we incubated the cells at 37 °C in 5% CO_2_ and added 1 mL of calcein AM (provided by the kit). After incubation, the medium was replaced by PBS and the cells were analyzed using a Leica TCS SPE confocal microscope (Leica, Buffalo Grove, IL, USA).

### 2.7. Doxorubicin Accumulation

Because of the pumping-out effect that P-gp has on cytotoxic moieties, compound 160a’s MDR reversal effect, and that of another P-gp inhibitor (verapamil), was studied using the corresponding increase in DOX (a P-gp substrate) accumulation in cells. The doxorubicin accumulation test was performed as previously reported [23], with minor modifications. Briefly, approximately 4 × 10^5^ cells from the parental cell lines/DOX-resistant cell lines (A549/DOX-resistant A549, KYSE150/DOX-resistant KYSE150, LCC6/LCC6MDR, and MCF-7/MX-100) were seeded into a 6-well cell culture plate and incubated with 20 μM DOX in an appropriate culture medium with different concentrations (10 μg/mL, 20 μg/mL, and 50 μg/mL) of compound 160a (34.12 μM, 68.24 μM, and 170.60 μM, respectively), 50 μg/mL (109.99 μM) of verapamil as a positive control, and 0.1% DMSO as a blank control. After 150 min of incubation at 37 °C in 5% CO_2_, the cells were washed with cold PBS and then lysed with lysis buffer (0.75 M HCl and 0.2% Triton-X100 in 2-propanol). The fluorescence level of DOX in the lysate was measured using a CLARIOstar microplate reader (BMG LABTECH, Ortenberg, Germany) with the excitation and emission wavelength pair of 460 and 587 nm, respectively. A working solution of DOX (0, 0.05, 0.1, 0.2, 0.5, 1, 2, and 5 μg/mL) was prepared, and a standard curve of DOX fluorescence signal versus DOX concentration was plotted to calculate the intracellular concentration of DOX. Finally, a graph of intracellular DOX concentration versus compound concentration was plotted to evaluate the MDR reversal effect of compound 160a compared to that of verapamil.

### 2.8. Duration of the MDR Reversal Effect

To evaluate the duration of compound 160a’s action as a modulator, approximately 5000 DOX-resistant cells (A549/DOX and MX100) were plated in each well of a flat-bottom 96-well plate in 100 μL of the appropriate culture medium. After the cells had settled down, the old culture medium was replaced by fresh medium with 10 μg/mL (34.12 μM) of compound 160a and the cells were incubated at 37 °C in 5% CO_2_ overnight. Then, we washed the cells with fresh culture medium to remove compound 160a, incubated them in fresh culture medium for 0, 0.5, 1, and 2 h, and added doxorubicin (0.15 μg/mL for A549/DOX cells; 0.05 μg/mL for MX100 cells). MTS working solution (100 μL) was added to each well after the culture medium’s removal at 0 h, 6 h, 24 h, 48 h, and 72 h, respectively, to determine cell viability, and the cells were then incubated at 37 °C in 5% CO_2_ in the dark for a period of time. Cell viability was then determined by measuring the absorbance of each well at 492 nm using a microplate reader (Bio-RAD, Ultrmark, Microplate Imaging System, USA). The relative growth (compared with the cell viability at 0 h) of each cell line was calculated using the equation:$$\frac{{\left[A\right]}_{T}}{{\left[A\right]}_{T0}},$$
where [A]_T_ is the absorbance at time point T and [A]_T0_ is the absorbance at 0 h. This assay was performed in triplicate [23].

## 3. Results

### 3.1. Molecular Docking Analysis

To identify the possible molecular targets for compound 160a, a molecular docking analysis was performed using the SEA platform (http://sea.bkslab.org). According to the manufacturer’s instructions, the probability that the compound will bind to the target protein is significant if the expected value (the *p*-value) is lower than 1 × 10^−10^. A target protein with a lower *p*-value possesses high affinity to the test compound. The maximum target complementary value (MaxTC) indicates the level of ligand–target complementarity. A higher MaxTC value suggests a more suitable binding to the target ligand. Table 1 shows compound 160a’s top three possible targets.

Multidrug resistance protein-1 (MDR-1, which is coded by the ABCB1 gene and also known as P-gp) possessed the second lowest expected value (3.24e^−18^) and the highest maximum target complementary value (0.46). Therefore, P-gp was predicted to be compound 160a’s most possible target and was further investigated in this study.

### 3.2. Cytotoxic Effect of Doxorubicin Combined with 160a

Doxorubicin (Dox) is one of the substrates for p-glycoprotein. Dox-resistant cell lines were developed by the prolonged culture of cells with doxorubicin (in a medium containing an increasing concentration of doxorubicin (0.1 μg/mL, 0.2 μg/mL, 0.5 μg/mL, 0.75 μg/mL, and 1.00 μg/mL). The cytotoxicity of doxorubicin was evaluated in parental and dox-resistant cell lines to check for the development of Dox resistance (Figure 2a,c). The Dox resistance effect of compound 160a was determined by co-culturing the Dox-resistant cells with different concentrations of doxorubicin using an MTS assay. The results are shown in Figure 2b,d. To evaluate compound 160a’s toxicity to non-cancer cells, NE-3 cells were treated with an increasing concentration of compound 160a or doxorubicin. The results are shown in Figure 2e,f. The IC_50_ (half maximal inhibitory concentration) value was determined as the concentration of tested compounds which showed 50% reduction of MTS signal compared with vehicle control. The IC_50_ values are summarized in Table 2.

According to Figure 2a,c, A549/DOX cells and KYSE150/DOX cells had a significantly higher resistance to doxorubicin than parental A549 cells. Figure 2b,d show that compound 160a significantly reversed the cytotoxic effect of doxorubicin in both A549/DOX and KYSE/DOX cells.

The CI_50_ value was defined as the concentration of compound 160a or doxorubicin that produced a 50% reduction in the MTS signal compared with the vehicle control. The CI_50_ value of compound 160a in NE-3 cells was 2.470 µg/mL, and the CI_50_ value of doxorubicin in NE-3 cells was 0.048 µg/mL. These results suggest that compound 160a was less toxic than doxorubicin to non-cancer cells.

### 3.3. Analysis of the Synergistic Effect of Compound 160a

Although MTS assay data from a previous study indicate that doxorubicin combined with compound 160a has a higher cytotoxic effect than doxorubicin alone, the synergistic effect of compound 160a on A549/DOX cells has yet to be evaluated by the combination index (CI) value via the Compusyn program [24].

Figure 3 shows the results. The cytotoxic effect of doxorubicin combined with compound 160a was found to be higher than that of either doxorubicin alone or compound 160a alone. The CI value was calculated using the cytotoxic effect of the combined use of doxorubicin and compound 160a at the ratio of 1:10 (doxorubicin dose: 1 µg/mL, compound 160a dose: 10 µg/mL, cell viability percentage: 0.23; doxorubicin dose: 0.5 µg/mL, compound 160a dose: 5 µg/mL, cell viability percentage: 0.31). Figure 3b shows that the CI values of the combination are all less than 1, indicating that, in combination with doxorubicin, compound 160a exerts a synergistic effect on A549/DOX cells.

### 3.4. Effect of Compound 160a on Cancer Cell Proliferation

The effect of compound 160a on the proliferation of doxorubicin-resistant cancer cells was assessed using an MTS assay on DOX-resistant cell lines. Doxorubicin-resistant cells were co-cultured with compound 160a, doxorubicin, or compound 160a combined with doxorubicin, and cell viability was determined by MTS reagent at 0, 6, 24, 48, and 72 h. Figure 4 shows the results.

The doxorubicin-resistant cancer cells in the control group were found to grow faster than doxorubicin-resistant cancer cells treated with either compound 160a, doxorubicin, or a combination of compound 160a and doxorubicin. The growth of the cancer cells that were treated with the combination of 160a and doxorubicin was found to be suppressed. Treatment with a combination of doxorubicin and compound 160a was found to have stronger suppressive effects on cancer cell growth than treatment with doxorubicin alone. These results indicate that compound 160a enhanced the cytotoxic effects of doxorubicin on DOX-resistant cancer cells.

### 3.5. Multidrug Resistance Analysis

To study compound 160a’s MDR reversal effect on p-glycoprotein, a multidrug resistance kit (Cayman) was employed to determine the intracellular accumulation of calcein AM (a substrate for p-glycoprotein) using flow cytometry and confocal microscopy. A vehicle control, 2 μg/mL, 10 μg/mL, and 20 μg/mL compound 160a, and 50 μg/mL verapamil (a p-glycoprotein inhibitor that was used as a positive control) were tested to study their effect on the intracellular accumulation of calcein AM, which can be evaluated by the intensity of green fluorescence. Figure 5 shows the percentage of calcein AM retention regulated mby compound 160a in the doxorubicin-resistant A549 cell line.

Figure 5a shows that the intracellular accumulation of calcein AM had a statistical average increase of 7.06% in the 20 μg/mL compound 160a-treated group (1.75 in the 2 μg/mL compound 160a-treated group and 3.78% in the 10 μg/mL compound 160a-treated group) compared with the vehicle control group.

To further study compound 160a’s p-glycoprotein-inhibiting effect, an SPE Confocal Microscope (Leica, Buffalo Grove, IL, USA) was used to observe the green fluorescence signal that was generated by the calcein AM in the doxorubicin-resistant A549 cell line after treatment with different concentrations of compound 160a (5 μg/mL, 25 μg/mL, and 50 μg/mL) for 30 min. Figure 6 shows the bright field, overlayer, and calcein AM signal images that were captured by the confocal microscope. The fluorescence intensity of the intracellular calcein AM in the A549 andA549/DOX cell lines was measured by software Image J and is summarized in Figure 7.

According to the confocal microscopy images, the fluorescence signal intensity that was generated by calcein AM showed an increasing trend after treatment with either an increasing concentration (5 μg/mL, 25 μg/mL, and 50 μg/mL) of compound 160a (the calcein AM fluorescence signal intensity showed a dose-dependent behavior) or verapamil in the doxorubicin-resistant A549 cell line. This result indicates that compound 160a could increase the amount of calcein AM that accumulates inside cancer cells by the inhibition of the pumping-out effect of p-glycoprotein.

### 3.6. Doxorubicin Accumulation Test

To provide additional evidence for compound 160a’s MDR reversal effect, a doxorubicin accumulation test was performed on the cell lines LCC6/LCC6 MDR, KYSE150/KYSE150-DOX, A549/A549-DOX, and MCF-7/MX-100. The cells were cultured in 20 µM doxorubicin, treated with different concentrations of compound 160a, and then lysed for direct spectrophotometric detection. Figure 8 shows the effect of compound 160a on intracellular doxorubicin accumulation in parental/DOX-resistant cells. The concentration of intracellular doxorubicin was calculated using a standard curve plot of fluorescence intensity versus concentration.

According to the results of the intracellular doxorubicin accumulation test, the DOX concentrations in parental cells were all significantly higher than those in the corresponding DOX-resistant cells, indicating that DOX-resistant cell lines could pump out doxorubicin as a result of the p-glycoprotein function. Additionally, in the DOX-resistant cell lines, the intracellular doxorubicin concentration increased in a dose-dependent manner with an increasing concentration of compound 160a, suggesting that compound 160a may play a role in reversing the MDR effect by inhibiting the p-glycoprotein function.

### 3.7. Potency of Reversal of the Function of p-Glycoprotein

To evaluate the potency of compound 160a’s MDR reversal effect, the dox-resistant cancer cell lines A549/DOX and MX100 were incubated with 10 µg/mL compound 160a overnight followed by the removal of compound 160a by washing and incubation with fresh culture medium for different periods of time (0, 0.5, 1, and 2 h) before doxorubicin treatment. Cell viability was determined by an MTS assay. The relative MTS value at different time points was calculated by the ratio of the MTS value at a time point (0.5, 1, or 2 h) to the MTS value at time 0. The results are shown in Figure 9.

Compound 160a’s potency for reversing the p-glycoprotein’s effect was found to be remarkable immediately after its removal, and the MDR reversal effect was still significant after 1 h in A549/DOX cells and after 0.5 h in MX100 cells. In contrast, an MDR reversal effect was not observed after 2 h from the removal of compound 160a. These results demonstrate that the MDR reversal effect of compound 160a can persist for at least 1 h.

## 4. Discussion

The use of the quinoline compound 160a to reverse p-glycoprotein-based MDR was investigated in this study. The compound 160a (8-(3-methoxybenzyloxy) quinoline-2-carbaldehyde) was first prepared in 2012 by Dr. Penny Chan Sau-hingin from our research group based on the structure–activity approach. The SEA program was used to predict the possible human protein targets for the compound 160a. P-glycoprotein possessed the highest maximum target complementary value and the second lowest expected value (3.235e^−18^). Moreover, overexpression of p-glycoprotein was shown to be one of the major mechanisms underlying MDR in cancers [8]. On the other hand, the predicted target with the lowest expected value (2.272e^−19^)—MTNR1A—encodes one of the two high-affinity forms of a receptor for melatonin. Diseases that are related to MTNR1A include idiopathic scoliosis and scoliosis [26]. Although MTNR1A is not directly associated with this study on identifying novel anticancer agents, it deserves further investigation in the future. p-glycoprotein was selected as the target of compound 160a in this study.

Doxorubicin has been used to treat many types of cancers, including leukemia and bladder, breast, and lung cancer [27], and is a well-known substrate for p-glycoprotein, which induces multidrug resistance. Drug resistance is the main limitation to doxorubicin therapy [28]. Therefore, doxorubicin was chosen in this study to be an indicator for the evaluation of the possible anti-MDR effect of compound 160a. In the present study, doxorubicin-resistant cells were developed using escalating doses of doxorubicin. The observed resistance effect was possibly due to the increased level of membrane surface p-glycoprotein as well as the inhibition of other kinases. To quantitatively measure the P-gp expression level, monoclonal antibodies, such as 4E3, UIC2, and JSB-1, can be used to detect P-gp by flow cytometry or immunohistochemistry [29]. Besides the P-gp-induced resistance effect, resistance caused by other mechanisms, including Rho-associated protein kinase 2, will be investigated in the future.

The cytotoxicity effect and the proliferation of cells were measured using an MTS assay. The change of MTS tetrazolium compound into a colored formazan product, which is accomplished by NADPH or NADH that is produced via dehydrogenase enzymes in metabolically active cells, indicates the degree of cell viability [30]. Hence, the MTS signal directly reflects the cell metabolism rather than the actual numbers of live cells. Assays for the evaluation of cell viability and proliferation include the Alamar blue assay, the neutral red uptake assay, the propidium iodide assay, the lactate dehydrogenase assay, the bromodeoxyuridine incorporation assay, and the 3H-thymidine incorporation assay [31]. These assays can be used to provide evidence of a compound effect of cytotoxicity on cell proliferation.

According to Figure 2, in both A549/DOX and KYSE150/DOX cells, doxorubicin showed higher cytotoxicity when combined with compound 160a (a low or a high dose). There are two possible reasons for this. One reason is that compound 160a is cytotoxic; the other reason is that compound 160a produces a synergistic effect when combined with doxorubicin via certain mechanisms. We hypothesized that compound 160a reversed the drug-resistance effect induced by p-glycoprotein and enhanced the intercellular concentration of doxorubicin to produce higher cytotoxicity. Therefore, the synergistic effect and its underlying mechanisms were investigated.

The synergistic effect of doxorubicin in the presence of compound 160a on A549/DOX cells was analyzed using the Compusyn program. The percentage of cell viability, as determined from the MTS value, was applied to evaluate the drug- and drug–drug interaction-related cytotoxicity. The CI values of the compound 160a and doxorubicin combination were all found to be less than 1. It has been reported that, after calculation using the Chou–Talalay model [24], a CI value <1, =1, and >1 indicates synergism, an additive effect, and antagonism, respectively. Therefore, we concluded that compound 160a exerted a synergistic effect in combination with doxorubicin on A549/DOX cells. Similarly, treatment with a combination of MEK and AKT inhibitors in human non-small-cell lung cancer was found to be more effective than treatment with each drug alone [32], and pre-treatment with propranolol before chemotherapy, such as 5-fluorouracil or paclitaxel, may increase relapse-free and overall survival in breast cancer patients [33]. The synergistic effects that were observed in the abovementioned studies were confirmed by the CI values according to the Chou–Talalay model.

The proliferation rates of A549/DOX, KYSE150/DOX, LCC6/MDR, and MX100 cells treated with compound 160a, doxorubicin, or a combination of compound 160a and doxorubicin were further investigated. In all four DOX-resistant cancer cell lines, the cells in the control group grew faster than those in the compound 160a, doxorubicin, and combination groups. The growth of cells that were treated with a combination of 160a and doxorubicin was significantly suppressed. Treatment with a combination of doxorubicin and compound 160a showed stronger suppressive effects on cell growth than treatment with doxorubicin alone after 72 h of incubation (A549/DOX cells after 6 h of incubation, MX100 cells after 48 h of incubation, and KYSE150/DOX cells and LCC6/MDR cells after 72 h of incubation). Thus, compound 160a significantly promoted the inhibitory effect of doxorubicin on cell proliferation. These results suggest that compound 160a can enhance the cytotoxic effects of doxorubicin on DOX-resistant cancer cells.

The ability of compound 160a to reverse the p-glycoprotein-based MDR in DOX-resistant cell lines was determined using flow cytometry, confocal microscopy, and a multi-drug resistance kit. Calcein AM was applied as the p-glycoprotein substrate instead of doxorubicin. Calcein AM is a cell-permeable non-fluorescent dye that exhibits strong green fluorescence once it has been transported into viable cells as a result of an intracellular esterases-mediated reaction [34]. Thus, calcein AM may be selected as a probe for identifying a p-glycoprotein modulation effect. Calcein AM can be transferred into cells directly or via the intracellular vesicles; therefore, the green fluorescence signal that can be detected by flow cytometry or observed using a confocal microscope consists of these two parts. Figure 5, Figure 6 and Figure 8 show that compound 160a reduced the overall intracellular signal. The percentage of calcein AM retention in the doxorubicin-resistant A549 cell line was found to have an increasing trend after treatment with either different concentrations of compound 160a or verapamil (a well-known p-glycoprotein inhibitor that, in this study, was applied as a positive control [35]). To further study the p-glycoprotein-inhibiting effect of compound 160a, an SPE Confocal Microscope was used to observe the green fluorescence signal that was generated by calcein AM in the doxorubicin-resistant A549 cell line after treatment with compound 160a. The intensity of the fluorescence signal that was generated by calcein AM had an increasing trend after treatment with compound 160a in the doxorubicin-resistant A549 cell line. The results of these two tests suggest that compound 160a can increase the intensity of the intracellular green fluorescence produced by calcein AM in DOX-resistant cell lines, indicating that compound 160a has the ability to reverse the multi-drug resistance conferred by p-glycoprotein by inhibiting its pumping-out effect.

To provide more evidence of compound 160a’s MDR reversal effect, a doxorubicin accumulation test was performed as previously described [23]. Cell membranes, including intracellular vesicles, were lysed to determine the total amount of intracellular doxorubicin. Figure 8 shows the effect of compound 160a on intracellular DOX accumulation in parental/DOX-resistant cells. DOX accumulation was greater in the parental cells, which contained a higher amount of DOX than the DOX-resistant cells. The intracellular DOX concentration was 1.75 times higher in the LCC6 cells than in the LCC6/MDR cells; 1.56 times higher in the KYSE150 cells than in the KYSE150/DOX cells; 1.97 times higher in the A549 cells than in the A549/DOX cells; and in MCF-7 cells it was 2.06 times higher than in MX100 cells in the absence of compound 160a, which may be due to the DOX efflux in the DOX-resistant cells. Additionally, in the DOX-resistant cell lines, the intracellular doxorubicin concentration was increased in a dose-dependent manner with an increasing concentration of compound 160a, suggesting that compound 160a may exert its MDR reversal effect by inhibiting p-glycoprotein’s function. When compared with the effect of verapamil, both 20 µg/mL and 50 µg/mL compound 160a enhanced the intracellular DOX accumulation to a greater degree than 50 µg/mL verapamil. In A549/DOX cells, 10 µg/mL of compound 160a had a stronger enhancing effect than verapamil with respect to DOX accumulation. In KYSE150/DOX and MX100 cells, 20 µg/mL and 50 µg/mL compound 160a exhibited a stronger promoting effect than verapamil with respect to DOX accumulation. These data suggest that compound 160a exhibits a stronger DOX-accumulation-promoting effect in some DOX-resistant cancer cell lines (KYSE150/DOX, A549/DOX, and MX100) than verapamil with the use of a lower dose. It was reported that a series of quinoline compounds (6,7-dimethoxy-2-(2-(4-(1H-1, 2, 3- triaxol-1-yl) phenyl) ethyl)-1, 2, 3, 4-tetrahydroisoquinolines) possess a potent MDR-reversal effect that could enhance the effect of other MDR-related cytotoxic agents and increase the accumulation of DOX in K562/A02 MDR cells [23].

Taken together, our results indicate that compound 160a exerts a synergistic effect in the presence of the p-glycoprotein substrate doxorubicin by reversing the drug efflux action of p-glycoprotein. The test of the duration of compound 160a’s MDR reversal effect showed that its potency for reversing p-glycoprotein-mediated MDR was present after its removal and remained significant after 1 h in A549/DOX cells and after 0.5 h in MX100 cells. In contrast, the MDR reversal effect was not observed 2 h after the removal of compound 160a. These results demonstrate that the MDR reversal effect on p-glycoprotein induced by compound 160a can persist for at least 1 h. Similar results were reported previously with the use of verapamil, whose MDR-reversing potency was found to decrease immediately after removal [23], and the use of cyclosporin A, which lost its MDR-reversing activity within 60 min after removal [36]. The potency assay for doxorubicin accumulation showed that the MDR-reversing potency of compound 160a persisted for approximately 0.5–1 h. This result suggests that a more appropriate administration scheme may be one in which treatment with anti-MDR drugs is performed first, and is then followed by treatment with other anticancer drugs after a definite period of time, in order to minimize drug–drug interaction and reverse MDR at the same time.

In this study, we demonstrated the ability of compound 160a to overcome drug resistance via inhibition of P-gp. A more appropriate model to study P-gp would be cell lines that have been engineered to over-express P-gp and selected for higher expression of P-gp by FACS sorting (fluorescence activated cell sorting). Vectors with P-glycoprotein or mock vectors can be transfected into the cells. The expression of P-gp at the cell surface can be determined with fluorescence-activated cell sorting after staining with one of the P-gp-specific monoclonal antibodies MRK-16 and UIC2 [37], which recognize the extracellular region of the protein. Then, the compound effect in P-gp-overexpressing cells can be assessed by a comparison with the mock-vector-transfected control cells.

The development of drug resistance during chemotherapy is a major obstacle in cancer therapy; hence, treatment with a combination of anti-cancer drugs and anti-MDR agents may be a desirable strategy [38]. Most drugs are metabolized by CYP450 enzymes in the liver, whose activities can be inhibited or induced by many kinds of drugs, resulting in drug–drug interactions when two or more drugs are combined and administrated at the same time that may cause serious adverse reactions or even toxicity [39].

Overall, the present study has comprehensively described the MDR-reversal effect of the quinoline compound 160a based on p-glycoprotein modulation. The results of this study show that the studied quinoline compounds have the potential to reverse P-gp-mediated MDR in anti-cancer treatments.

## 5. Conclusions

The quinoline-based compound 160a, in combination with doxorubicin, exerts a synergistic effect on DOX-resistant cancer cells. The compound 160a probably reverses MDR by modulating p-glycoprotein-based drug efflux. The strong and persistent MDR reversal effect suggests that compound 160a has potential as an anti-MDR drug and could be combined with anti-cancer drugs to produce a greater therapeutic effect and lower toxicity.

## Figures and Tables

**Figure 1 biology-08-00075-f001:**
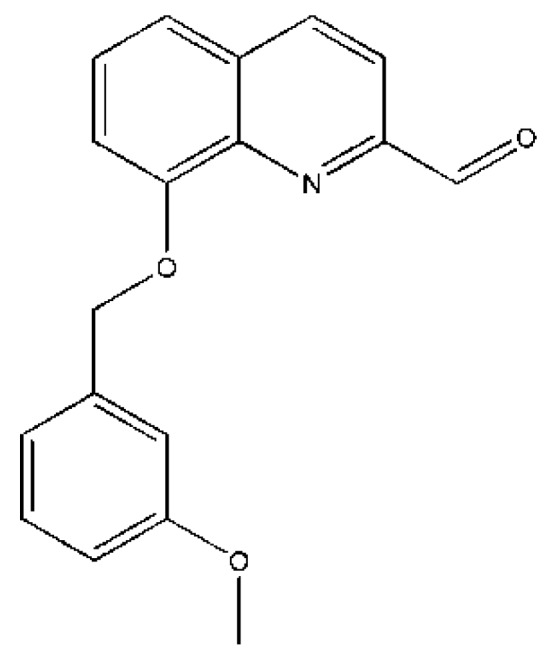
The structure of compound 160a.

**Figure 2 biology-08-00075-f002:**
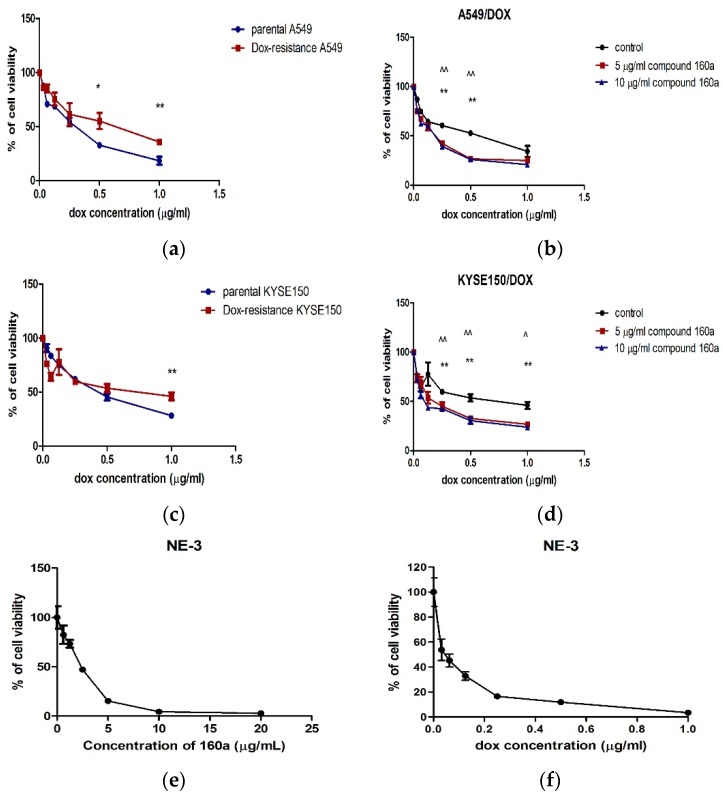
Cytotoxic effect of different concentrations of doxorubicin on parental and doxorubicin-resistant cancer cells co-cultured with compound 160a. (**a**) The relative MTS value of doxorubicin in the parental and dox-resistant A549 cell lines; (**b**) the relative MTS value of doxorubicin combined with compound 160a (5 µg/mL, 10 µg/mL) in the dox-resistant A549 cell lines; (**c**) the relative MTS value of doxorubicin in the parental and dox-resistant KYSE150 cell lines; (**d**) the relative MTS value of doxorubicin combined with compound 160a (5 µg/mL, 10 µg/mL) in dox-resistant KYSE150 cells; (**e**) the relative MTS value of compound 160a in NE-3 non-cancer cells; (**f**) the relative MTS value of doxorubicin in NE-3 non-cancer cells. Dissolved in dimethyl sulfoxide (DMSO) (0.1%) was applied as a vehicle control (*n* = 4). ^ *p* < 0.05, ^^ *p* < 0.01 comparing the 5 µg/mL compound 160a group with the control group; * *p* < 0.05, ** *p* < 0.01 comparing the 10 µg/mL compound 160a group with the control group.

**Figure 3 biology-08-00075-f003:**
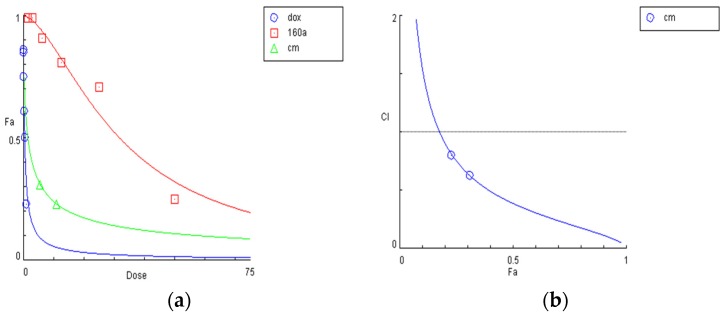
Analysis of the synergistic effect of compound 160a and doxorubicin on the dox-resistant cancer cell line A549/DOX using the CompuSyn program. (**a**) Dose–effect curve of compound 160a, doxorubicin, and compound 160a combined with doxorubicin versus relative MTS value; (**b**) Fa–CI plot (Chou–Talalay Plot). Fa, fraction of system affected; CI, combination index. CI = 1 indicates an additive effect; CI < 1 indicates synergism; CI > 1 indicates antagonism [25]. *n* = 3. Cm, combination; Dox, doxorubicin.

**Figure 4 biology-08-00075-f004:**
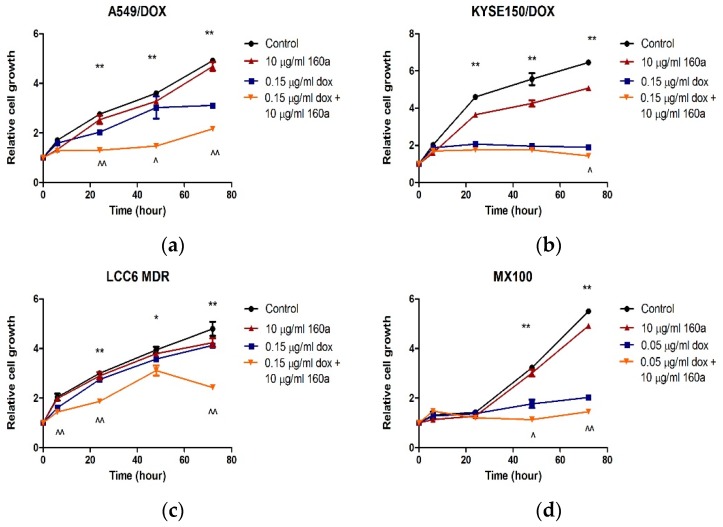
Cell proliferation curves of doxorubicin-resistant cells co-cultured with compound 160a, doxorubicin, and compound 160a combined with doxorubicin. (**a**) A549/DOX cells; (**b**) KYSE150/DOX cells; (**c**) LCC6/MDR cells; (**d**) MX100 cells. DMSO (0.1%) was applied as a vehicle control. *n* = 3. * *p* < 0.05, ** *p* < 0.01 comparing the doxorubicin combined with compound 160a group with the control group; ^ *p* < 0.05, ^^ *p* < 0.01 comparing the doxorubicin combined with compound 160a groups with the doxorubicin alone group.

**Figure 5 biology-08-00075-f005:**
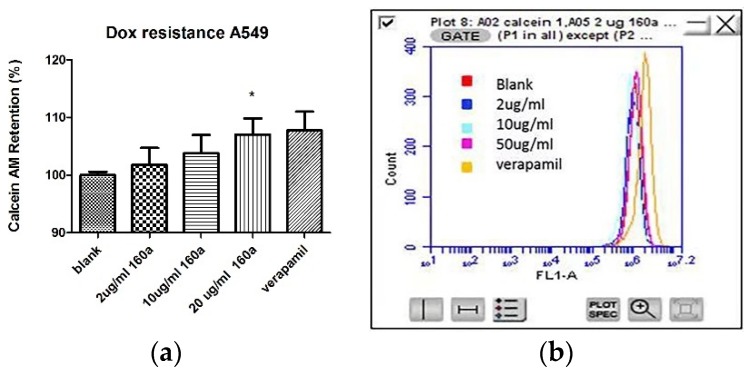
Effects of reversing the MDR phenotype in the doxorubicin-resistant A549 cell line. (**a**) Calcein acetoxymethyl (AM) retention produced by compound 160a on the doxorubicin-resistant A549 cell line; (**b**) the intensity of the green fluorescence signal from the calcein AM counted in the filter channel (FL-1) using a flow cytometric analysis of calcein AM with an increasing concentration of compound 160a (2, 10, and 20 μg/mL). DMSO (0.1%) was applied as a vehicle control. Verapamil (50 μg/mL) was applied as a positive control. *n* = 3. * *p* < 0.05, comparing with blank control group.

**Figure 6 biology-08-00075-f006:**
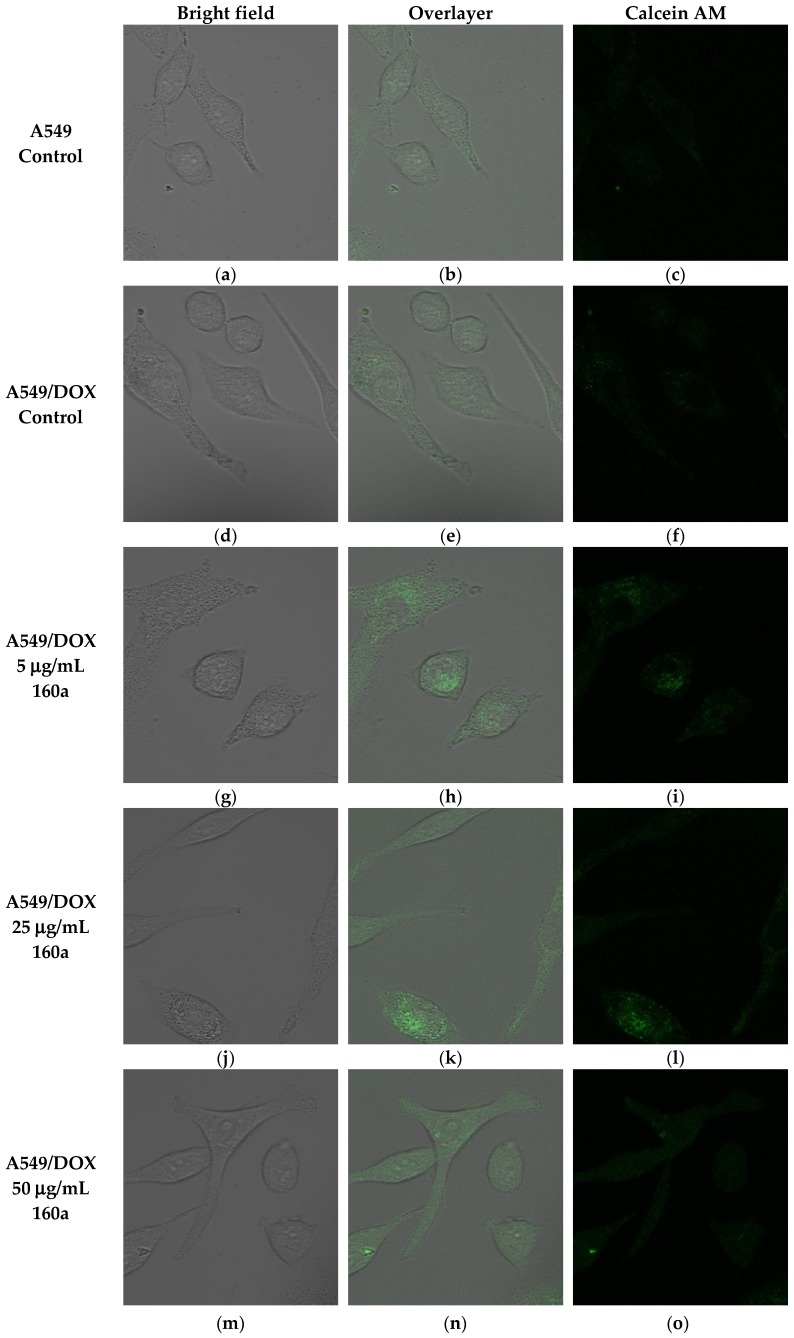
Images of fluorescent signals of calcein AM accumulated in doxorubicin (dox)-resistant A549 cells after treatment with different concentrations of compound 160a for 30 min. (**a**–**c**): Calcein AM signals of A549 parental cells treated with DMSO; (**d**–**f**): Calcein AM signals of dox-resistant A549 cells treated with DMSO; (**g**–**i**): Calcein AM signals of dox-resistant A549 cells treated with 5 μg/mL compound 160a; (**j**–**l**): Calcein AM signals of dox-resistant A549 cells treated with 25 μg/mL compound 160a; (**m**–**o**): Calcein AM signals of dox-resistant A549 cells treated with 50 μg/mL compound 160a; (**p**–**r**): Calcein AM signals of dox-resistant A549 cells treated with verapamil. DMSO (0.1%) was applied as a vehicle control; 50 μg/mL verapamil was applied as a positive control.

**Figure 7 biology-08-00075-f007:**
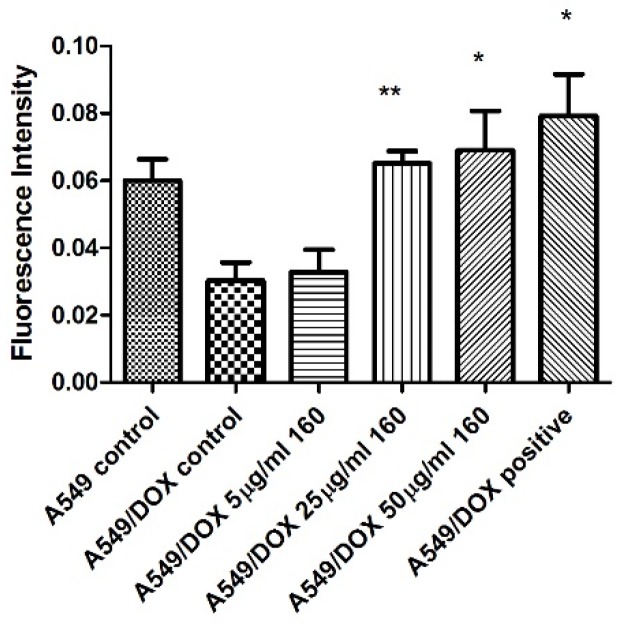
Summary of the fluorescence intensity of intracellular calcein AM in the A549 and A549/DOX cell lines treated with compound 160a and observed using a confocal microscope. DMSO (0.1%) was applied as a vehicle control; 50 μg/mL verapamil was applied as a positive control. Fluorescence intensity was measured by Image J.

**Figure 8 biology-08-00075-f008:**
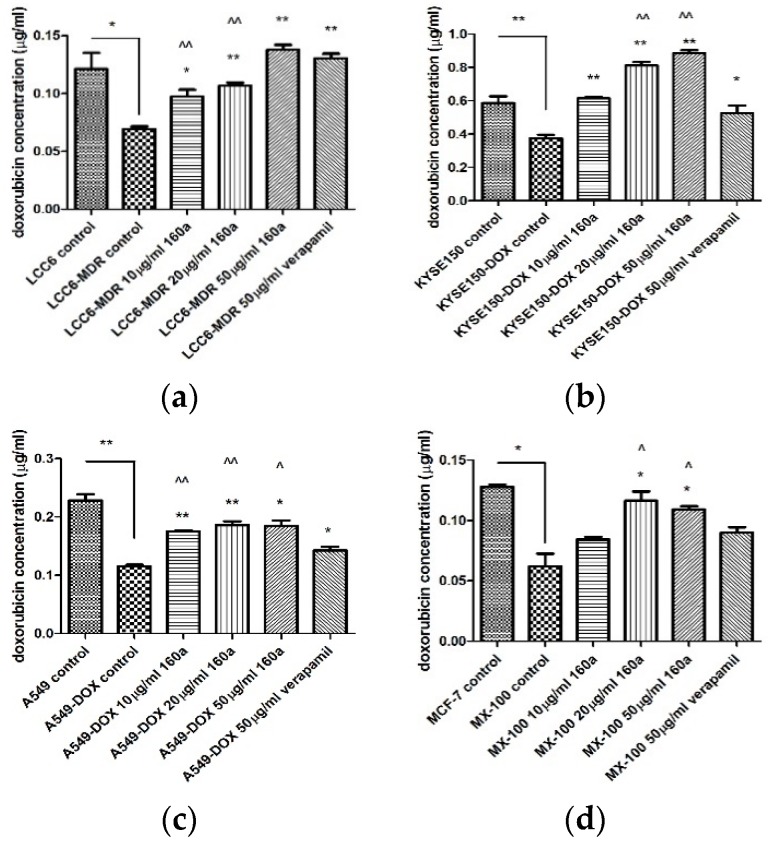
Effect of compound 160a on intracellular DOX accumulation in parental/DOX-resistant cells. (**a**) Intracellular doxorubicin concentration of LCC6/LCC6 MDR cells treated with compound 160a; (**b**) Intracellular doxorubicin concentration of KYSE150/KYSE150-DOX cells treated with compound 160a; (**c**) Intracellular doxorubicin concentration of A549/A549-DOX cells treated with compound 160a; and (**d**) Intracellular doxorubicin concentration of MCF-7/MX-100 cells treated with compound 160a. DMSO (0.1%) was applied as a negative control; 50 µg/mL verapamil was applied as a positive control. *n* = 3. * *p*
< 0.05, ** *p*
< 0.01 comparing the other groups with the dox-resistant control group; ^ *p* < 0.05, ^^ *p* < 0.01 comparing the compound-160a-treated groups with the verapamil group.

**Figure 9 biology-08-00075-f009:**
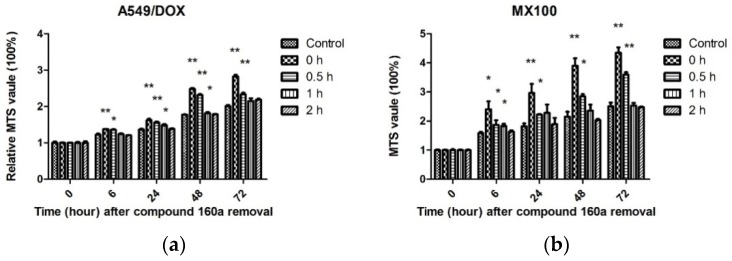
Duration of the MDR reversal effect of compound 160a on doxorubicin in DOX-resistant cancer cells. (**a**) Duration of the MDR reversal effect of compound 160a on doxorubicin in A549/DOX cancer cells; (**b**) Duration of the MDR reversal effect of compound 160a on doxorubicin in MX100 cancer cells. DMSO (0.1%) was applied as a vehicle control. Cell viability was determined by an MTS assay. *n* = 3. * *p* < 0.05; ** *p* < 0.01.

**Table 1 biology-08-00075-t001:** Top three predicted human protein targets for compound 160a based on the similarity ensemble approach (SEA).

Rank	Target Key	Target Name	Description	*p*-Value	MaxTC
1	MTR1A_HUMAN+5	MTNR1A	Melatonin receptor type 1A	2.272e^−19^	0.40
2	MDR1_HUMAN+5	ABCB1	Multidrug resistance protein 1	3.235e^−18^	0.46
3	ROCK2_HUMAN+5	ROCK2	Rho-associated protein kinase 2	3.902e^−18^	0.32

MaxTC, maximum target complementary value.

**Table 2 biology-08-00075-t002:** IC_50_ values (μg/mL) of doxorubicin (Dox), Dox combined with compound 160a on parental cancer cell lines or Dox-resistance cancer cell lines, and Dox or compound 160a alone on non-tumor cell line (NE-3).

Cell Lines	IC_50_ Value (μg/mL)		
	Dox	Dox + 5μg/mL 160a	Dox + 10μg/mL 160a
Parental A549	0.408	N.A.	N.A.
Dox-resistance A549	0.542	0.182	0.195
Parental KYSE150	0.475	N.A.	N.A.
Dox-resistance KYSE150	0.735	0.164	0.057
	**Dox**	**160a**	
NE-3	0.048	2.470	

N.A. not applicable. Results were calculated by GraphPad Nonlinear regression analysis.

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
