# Peer review of "Development of a Novel Quinoline Derivative as a P-Glycoprotein Inhibitor to Reverse Multidrug Resistance in Cancer Cells"

_biology, 2019, doi:10.3390/biology8040075_

Round 1

Reviewer 1 Report

This work evaluates the putative multidrug resistance reversal activity of the compound 160a. In my personal opinion the work has scientific relevance and is well planed, however the English needs to be reviewed and there are some minor changes that need to be done before publication.

Materials and methods; the synthesis of compound 160a needs to be described with more detail or a bibliographic reference where we can find the description of the compound synthesis. Line 93; the cell lines were not examined in the study; they were used to evaluate the effect of the drug. In the same paragraph the word one before the name of each cell line should be replaced by the. Line 105 a reference describing the process to select a doxorubicin resistant cell line should be introduced in the text. Another bibliographic reference for cell maintenance at the end of the text. Cytotoxicity assay – reference is missing. Line 145, the results are not recorded using a microplate reader…I believe that the author means that the results were analised by measuring the absorbance at 492nm (microplate reader BIO-Rad…)   Cell proliferation assay reference is missing Results Figure 2 – in the graphs instead of MTS value should be % of cell viability. Instead of MTS50 the statistical term should be use (IC50). The IC50 values are very important in this type of analysis, thus the author should include a table with the IC50 values. Figure 4 in the graphs instead of relative MTS activity, relative cell growth

Author Response

Point 1: Structure information should be provided for compound 160a, such as UPLC/MS/NMR.

Response 1: The MS figure, NMR figure and data are now available at the Supplementary Information.

Point 2: Molecular docking analysis showed that P-gp was predicted to be the binding target for 160a. The molecular modeling mode for 160a in P-gp should be provided.

Response 2: The molecular modelling mode for 160a in P-gp information has been offered in the Supplementary Information.

Point 3: P-value for Figure 5 should be calculated and added.

Response 3: The p-value of test group comparing with control group have been calculated, and * has been added to figure 5 accordingly.

Point 4: Color-coded curves for different drugs/concentration would be preferred in Figure 2 and Figure 4.

Response 4: Figure 2 and Figure 4 have been replaced to color-coded version in manuscript and figure file.

Reviewer 2 Report

In this manuscript, authors reported a novel quinoline derivative 160a, which was predicted to target p-glycoprotein and effectively reversed the multidrug resistance in cancer cells. The manuscript was well written with overall acceptable scientific quality, and could be considered for publication with minor revisions.

The comments are as following:

Structure information should be provided for compound 160a, such as UPLC/MS/NMR. Molecular docking analysis showed that P-gp was predicted to be the binding target for 160a. The molecular modeling mode for 160a in P-gp should be provided. P-value for Figure 5 should be calculated and added. Color-coded curves for different drugs/concentration would be preferred in Figure 2 and Figure 4.

Author Response

Point 1: the English needs to be reviewed and there are some minor changes that need to be done before publication.

Response 1: the manuscript has been edited by MDPI English editing service.

Point 2: the synthesis of compound 160a needs to be described with more detail or a bibliographic reference where we can find the description of the compound synthesis.

Response 2: The synthesis of compound 160a was described by the following added sentences at Section 2.1 Synthesis of Compound 160a:

Compound 160a was prepared by oxidation of 8-(3-methoxybenzyloxy)-2-methylquinoline with addition of selenium dioxide, pre-dried 1,4-dioxane and water in reflux for 24 hours. 8-(3-Methoxybenzyloxy)-2-methylquinoline was synthesized by nucleophilic substitution of commercially available 2-methyl-8-quinolinol with 3-methoxybenzyl bromide in DMF at room temperature.

Point 3: Line 93; the cell lines were not examined in the study; they were used to evaluate the effect of the drug.

Response 3: The text has been amended to “A total of nine cell lines were used to evaluate for the effect of the test compounds in this study.”

Point 4: In the same paragraph the word one before the name of each cell line should be replaced by the.

Response 4: All “one” before cell line have changed to “the”.

Point 5: Line 105 a reference describing the process to select a doxorubicin resistant cell line should be introduced in the text.

Response 5: A reference has been added to line 110.

Point 6: Another bibliographic reference for cell maintenance at the end of the text. Cytotoxicity assay – reference is missing.

Response 6: A reference has been added to line 125 for cell maintenance. A reference has been added to line 140 for cytotoxicity assay

Point 7: Line 145, the results are not recorded using a microplate reader…I believe that the author means that the results were analysed by measuring the absorbance at 492nm (microplate reader BIO-Rad…)

Response 7: The expression has been changed to “The results were analysed by measuring the absorbance at 492 nm using a microplate reader (Bio-RAD, Ultrmark, Microplate Imaging System, USA) to determine cell viability.”

Point 8: Cell proliferation assay reference is missing

Response 8:  A reference has been added to line 158.

Point 9: Results Figure 2 – in the graphs instead of MTS value should be % of cell viability. Instead of MTS50 the statistical term should be use (IC50). The IC50 values are very important in this type of analysis, thus the author should include a table with the IC50 values. Figure 4 in the graphs instead of relative MTS activity, relative cell growth

Response 9: “Relatively MTS value”in Figure 2 has changed to “% of cell viability. “Relative MTS activity” in Figure 4 has changed to “relative cell growth”.  The term MTS50 has changed to IC50 in the manuscript. The IC50 values were summarized in Table 2.

Table 2. IC50 values (μg/mL) of doxorubicin (Dox), Dox combined with compound 160a on parental cancer cell lines or Dox-resistance cancer cell lines, and Dox or compound 160a alone on non-tumor cell line (NE-3).

Cell lines

IC50 value (μg/mL)

Dox

Dox + 5μg/mL 160a

Dox + 10μg/mL 160a

Parental A549

0.408

N.A.

N.A.

Dox-resistance A549

0.542

0.182

0.195

Parental KYSE150

0.475

N.A.

N.A.

Dox-resistance KYSE150

0.735

0.164

0.057

Dox

160a

NE-3

0.048

2.470

N.A. not applicable. Results were calculated by GraphPad Nonlinear regression analysis from four parallel experiments.

This manuscript is a resubmission of an earlier submission. The following is a list of the peer review reports and author responses from that submission.

Round 1

Reviewer 1 Report

In their manuscript „Development of a Novel Quinoline Derivative 160a as a p-glycoprotein Inhibitor to Reverse Multi-Drug Resistance in Cancer Cells“ Zhou et al. introduce a new quinolone-based compound called 160a. They provide evidence that 160a is able to synergistically increase susceptibility of cancer cell lines towards doxorubicin, that have been made resistant towards this anti-cancer drug. By performing in silico analyses the authors identify the ABC transporter p-glycoprotein, an efflux pump  involved in cancer multi-drug resistance (MDR), as possible molecular target of 160a. Indeed, the authors are able to show increased accumulation of doxorubicin in cancer cell lines in the presence of 160a, indicating a possible inhibitory effect of this compound on p-glycoprotein.

Thus, this study proposes 160a as a novel promising p-glycoprotein inhibitor and anti-MDR agent for cancer therapy.

Overall the study is well designed and experiments performed are suitable to answer the questions asked. However, in several instances the conclusions drawn are not fully supported by the experimental data.

Specific points:

1) The development of new anti-cancer therapies should aim for high efficacy with minimal side effects, as stated by the authors in the introduction of the manuscript. However, no information or experiments are shown, where 160a is analyzed for its toxicity agains non-cancer cells. This needs to be addressed eg. by showing the IC50 of 160a for PBMCs from healthy donors.

2) Figure 4, middle of page 9 „The  growth  of  the  cancer  cells  with  the treatment  of  either  doxorubicin or 160a was suppressed, whereas the treatment of doxorubicin combined with compound 160a showed stronger suppressive effects on cell  growth. The results indicated that compound 160a enhanced the cytotoxic effects of doxorubicin on DOX-resistance cancer cells.“ and page 15, 2nd paragraph: „The  growth of cells with the treatment of either doxorubicin or 160a was suppressed,  whereas the treatment of doxorubicin combined with compound 160a showed stronger suppressive effects on cell growth.“

No statistical test is shown for doxorubicin or 160a alone vs. doxorubicin combined with compound 160a. Especially for MX100 and KYSE150/DOX these statements seem not to be correct.

3) Figure 5a: No statistics are mentioned.

4) End of page 15. „[…] which can be ascribed to p-glycoprotein mediated DOX efflux.“ This is not formally shown in this manuscript. Statement needs to be rephrased.

5a) Especially earlier p-gp inhibitors lacked sufficient affinity for their target. Thus high doses had to be used that resulted in unacceptable toxicities in in vivo tests. Here, the authors compare effective concentrations of 160a with that of verapamil as a positive control. Instead of a comparison of the concentrations, comparing the molarities of the compounds, at which a significant effect is observed, would be more relevant. - end of page 15

5b) End of page 15, start of page 16: „On LCC6/MDR, KYSE150/DOX, and A549/DOX  cells, even 10 μg/ml of compound 160a revealed a stronger enhancing effect than verapamil in DOX accumulation.“ No statistical significance information is provided for verapamil vs. 160a. Especially for LCC6/MDR this statement seems not to be correct.

The English language need to be corrected in several instances. Some, but not all necessary corrections are marked in the file attached.

Minor corrections:

End of page 12: A figure 4.31 is mentioned which is not found in the manuscript.

Author Response

Point 1: The development of new anti-cancer therapies should aim for high efficacy with minimal side effects, as stated by the authors in the introduction of the manuscript. However, no information or experiments are shown, where 160a is analyzed for its toxicity against non-cancer cells. This needs to be addressed eg. by showing the IC50 of 160a for PBMCs from healthy donors.

Response 1: The cytotoxicity test results of compound 160a and doxorubicin for non-cancer cells NE3 were added in Section3.2 as Figure 2 (e) and (f).

Point 2:  Figure 4, middle of page 9, The  growth  of  the  cancer  cells  with  the treatment  of  either  doxorubicin or 160a was suppressed, whereas the treatment of doxorubicin combined with compound 160a showed stronger suppressive effects on cell  growth. The results indicated that compound 160a enhanced the cytotoxic effects of doxorubicin on DOX-resistance cancer cells.“ and page 15, 2nd paragraph: „The  growth of cells with the treatment of either doxorubicin or 160a was suppressed,  whereas the treatment of doxorubicin combined with compound 160a showed stronger suppressive effects on cell growth.“

No statistical test is shown for doxorubicin or 160a alone vs. doxorubicin combined with compound 160a. Especially for MX100 and KYSE150/DOX these statements seem not to be correct.

Response 2: The statistical significance information of doxorubicin alone comparing with doxorubicin combined with compound 160a was provided to Figure 4. The describe statement was also amended as suggested in Results and Discussion sections.

Point 3: Figure 5a: No statistics are mentioned.

Response 3: It showed an increase of statistical average by 7.06% in 20 μg/mL compound 160a treated group (1.75 in 2 μg/mL compound 160a treated group, 3.78 % in 10 μg/mL compound 160a treated group) than vehicle control group.

This description was added to Section 3.5, under Figure 5.

Point 4: End of page 15. „[…] which can be ascribed to p-glycoprotein mediated DOX efflux.“ This is not formally shown in this manuscript. Statement needs to be rephrased.

Response 4: Statement has been rephrased accordingly to “which can be due to the DOX efflux in the DOX-resistance cells.”

Point 5: Especially earlier p-gp inhibitors lacked sufficient affinity for their target. Thus high doses had to be used that resulted in unacceptable toxicities in in vivo tests. Here, the authors compare effective concentrations of 160a with that of verapamil as a positive control. Instead of a comparison of the concentrations, comparing the molarities of the compounds, at which a significant effect is observed, would be more relevant. - end of page 15

Response 5:  The conversion of the doses of 160a and verapamil to molarity is added to the Materials and Methods section.

Point 6: End of page 15, start of page 16: „On LCC6/MDR, KYSE150/DOX, and A549/DOX  cells, even 10 μg/ml of compound 160a revealed a stronger enhancing effect than verapamil in DOX accumulation.“ No statistical significance information is provided for verapamil vs. 160a. Especially for LCC6/MDR this statement seems not to be correct.

Response 6: The statistical significance information of compound treated groups against verapamil groups was provided to Figure 8. The describe statement was also amended as suggested in Discussion section.

Point 7: End of page 12: A figure 4.31 is mentioned which is not found in the manuscript.

Response 7: This Figure was mentioned by mistake. We deleted this sentence in manuscript.

Special point: English language and style has been edited by MDPI.

Reviewer 2 Report

Overall Comments and Questions:
This article provides data on compound 160a reporting it’s activity as a P-gp inhibitor, capable of reversing multi-drug resistance in cancer cells.

1. Based on the molecular docking analysis, 3 targets were predicted, including P-gp. It isn’t clear how the compound is functioning as an inhibitor and further details of the docking analysis as well as any other additional analysis such as a functional kinase inhibition profiling would be more reliable and needs to be included to address other targets that may also be implicated in drug resistance.

2. The cell lines used appear to have been derived by selection in escalating doses of Doxoribicin. This is known to select for various mechanisms, and there are several concerns about using cell lines derived in this manner. Firstly, the level of surface Pgp can be variable but since it can be quantified by flow cytometry, this should be done to demonstrate that the resistant lines do in fact have higher levels of P-gp to the parentals. Secondly, the effects on P-gp specifically compared to other mechanisms of resistance, in particular those that could be targeted by the compound need to be also assessed and discussed. Since compound 160a can “target” ROCK2 (Table 1), which has been implicated in cell phenotype mediated mechanisms of drug resistance, the effects of the compound on ROCK activity and drug resistance also need to be assessed and compared.

3. Although the activity has been demonstrated in relation to P-gp, there is only associated links between the various experiments shown that link the ability of the compound to overcome resistance with inhibition of P-gp directly. A more appropriate model to assess the effects of the compound on P-gp specifically, would be cell lines engineered to over-express P-gp and selected for high(er) expression of P-gp by FACS sorting, to enable assessment of the compound in the P-gp high(er) expressing sub-lines compared to vector controls.

4. To demonstrate the ability of compound 160a to overcome multi-drug resistance, it would be useful to also assess the ability to overcome resistance to other drugs that P-gp can confer resistance to in the same assays.

5. There are several concerns in relation to the assays used for the experiments. The MTS assay has been used to assess cytotoxity (2.4) at 48 hrs and cell proliferation (2.5) across multiple time-points including 48 hrs. MTS is solution used for a colorimetric assays used for assessing cell number based on metabolically active cells. There are some limitations with this assay as it can sometimes provide readouts that reflect changes in cell metabolism rather than just cell number. Additionally, it isn’t clear how using MTS to quantify cell number can be used to measure cell proliferation and cell cytoxicity. There are several assays that would be more appropriate that measure cell viability/death and others that can distinguish between cell proliferation and cell death. To accurately assess and conclude about cytotoxicity, additional/different assays would need to be included.

6. P-gp has been detected in the membranes of intracellular vesicles and linked to drug sequestration into intracellular compartments. The effects of compound 160a need to be assessed and discussed in relation to this, particularly for the results shown in Figures 5-8.

Specific Points:

The molecular docking analysis prediction should be included in a Supplementary figure to support the details shown in Table 1.

The figures showing data need to provide specific details of the replicates used for the statistical analysis eg do the N=3 represent 3 technical replicates in one experiment or 3 experimental replicates each with multiple technical replicates (which is more reliable).

A control such as Verapamil (used in Fig 7/8) should also be included in other assays to compare the activity with compound 160a. The doses and time-points used in the various assays also need to be explained/justified for comparisons.

Author Response

Point 1: Based on the molecular docking analysis, 3 targets were predicted, including P-gp. It isn’t clear how the compound is functioning as an inhibitor and further details of the docking analysis as well as any other additional analysis such as a functional kinase inhibition profiling would be more reliable and needs to be included to address other targets that may also be implicated in drug resistance.

Response 1: The investigation on the roles of 160a in kinase inhibition is now ongoing that includes the kinase inhibition profiling and the overall findings will be soon reported in our next paper.

Point 2: The cell lines used appear to have been derived by selection in escalating doses of Doxoribicin. This is known to select for various mechanisms, and there are several concerns about using cell lines derived in this manner. Firstly, the level of surface Pgp can be variable but since it can be quantified by flow cytometry, this should be done to demonstrate that the resistant lines do in fact have higher levels of P-gp to the parentals. Secondly, the effects on P-gp specifically compared to other mechanisms of resistance, in particular those that could be targeted by the compound need to be also assessed and discussed. Since compound 160a can “target” ROCK2 (Table 1), which has been implicated in cell phenotype mediated mechanisms of drug resistance, the effects of the compound on ROCK activity and drug resistance also need to be assessed and compared.

Response 2: The issues about possible changes in P-gp in cancer cells has been discussed in the Discussion section paragraph 2 with the suggested experiment to be done in the next step. The study about the effects of the compound on ROCK activity and drug resistance is now ongoing and the overall findings will be soon reported in our next paper.

Point 3: Although the activity has been demonstrated in relation to P-gp, there is only associated links between the various experiments shown that link the ability of the compound to overcome resistance with inhibition of P-gp directly. A more appropriate model to assess the effects of the compound on P-gp specifically, would be cell lines engineered to over-express P-gp and selected for high(er) expression of P-gp by FACS sorting, to enable assessment of the compound in the P-gp high(er) expressing sub-lines compared to vector controls.

Response 3: We have added this view in the paragraph 10 of the Discussion section to provide another way to prepare P-gp over-expressed cell line to study the effect of 160a on P-gp.

Point 4: To demonstrate the ability of compound 160a to overcome multi-drug resistance, it would be useful to also assess the ability to overcome resistance to other drugs that P-gp can confer resistance to in the same assays.

Response 4: We are investigating the reversal of multi-drug resistance by compound 160a in breast cancer against paclitaxel which is the well-known target for P-gp. The overall results will be reported in another ongoing paper.

Point 5: There are several concerns in relation to the assays used for the experiments. The MTS assay has been used to assess cytotoxity (2.4) at 48 hrs and cell proliferation (2.5) across multiple time-points including 48 hrs. MTS is solution used for a colorimetric assays used for assessing cell number based on metabolically active cells. There are some limitations with this assay as it can sometimes provide readouts that reflect changes in cell metabolism rather than just cell number. Additionally, it isn’t clear how using MTS to quantify cell number can be used to measure cell proliferation and cell cytoxicity. There are several assays that would be more appropriate that measure cell viability/death and others that can distinguish between cell proliferation and cell death. To accurately assess and conclude about cytotoxicity, additional/different assays would need to be included.

Response 5: Other assays to determine cytotoxicity were discussed in the paragraph 3 of the Discussion section.

Point 6: P-gp has been detected in the membranes of intracellular vesicles and linked to drug sequestration into intracellular compartments. The effects of compound 160a need to be assessed and discussed in relation to this, particularly for the results shown in Figures 5-8.

Response 6:  This view was added in the paragraph 7 and 8 of the Discussion section.

Point 7: The molecular docking analysis prediction should be included in a Supplementary figure to support the details shown in Table 1.

Response 7: The simulation pictures of molecular docking for compound 160a, doxorubicin, or verapamil bound to P-gp were offered as the Supplementary materials.

Point 8: The figures showing data need to provide specific details of the replicates used for the statistical analysis eg do the N=3 represent 3 technical replicates in one experiment or 3 experimental replicates each with multiple technical replicates (which is more reliable).

Response 8: the experiment was done in triplicate under the same experimental condition.

Point 9: A control such as Verapamil (used in Fig 7/8) should also be included in other assays to compare the activity with compound 160a. The doses and time-points used in the various assays also need to be explained/justified for comparisons.

Response 9: In order to show the comparison using DOX with or without 160a in different cell lines (Fig. 8) after a series of basic assays, we subsequently used verapamil as the positive control as the overall comparison in the final part of the DOX-accumulation assays.  In the next series of reports for studying 160a and its novel derivatives, a series of known P-gp inhibitors (e.g. cyclosporine and erythromycin) will be used for comparison.

The doses and time-points used were added in Method and Materials section or Discussion Section.

Special point: English language and style has been edited by MDPI.

Round 2

Reviewer 1 Report

The authors have adequately addressed my points of critique in their revised manuscript in most cases.

One issue remains to be clarified pertaining to point 1/ answer 1 of my previous report:

The authors now have analyzed the toxicity of compound 160a on the non-cancer esophageal epithelial cell line NE-3 and found that the MTS50 value for this compound was 2.47 μg/mL (page 8 of the revised manuscript). Despite the authors' statement that "compound 160a was less toxic than doxorubicin to non-cancer cells" this raises major concern as the concentrations of compound 160a used to analyze its cytotoxicity on cancer cells were 5μg/ml or 10μg/ml (together with doxorubicin). At these concentrations the MTS values for non cancer cells are already below 20%. This might greatly hamper the prospect of compound 160a in its current form to be incorporated into a new cancer treatment regime. This needs at least to be critically discussed.

Reviewer 2 Report

All the points raised in the initial review need to be adequately addressed in the materials, methods and results section of this paper.

The queries and additional data requested need to be added to this paper as they are necessary to support the conclusions being made in this article,

The limitations can not be merely referred to in the discussion section, since they can be addressed in the experimental work being presented.